# The Medication Experience of TB/HIV Coinfected Patients: Qualitative Study

**DOI:** 10.3390/ijerph192215153

**Published:** 2022-11-17

**Authors:** Natália Helena de Resende, Ursula Carolina de Morais Martins, Djenane Ramalho-de-Oliveira, Dirce Inês da Silva, Silvana Spíndola de Miranda, Adriano Max Moreira Reis, Wânia da Silva Carvalho, Simone de Araújo Medina Mendonça

**Affiliations:** 1Graduate Program in Medicines and Pharmaceutical Assistance, College of Pharmacy, Universidade Federal de Minas Gerais, Belo Horizonte 31270-901, Brazil; 2Hospital Foundation of the State of Minas Gerais/Eduardo de Menezes Hospital, Belo Horizonte 30622-020, Brazil; 3Faculty of Medicine, Universidade Federal de Minas Gerais, Belo Horizonte 30130-100, Brazil

**Keywords:** HIV/AIDS, tuberculosis, medication experience, medication use, Brazil

## Abstract

Tuberculosis (TB) and human immunodeficiency virus/acquired immunodeficiency syndrome (HIV/AIDS) pharmacotherapy and the stigma related to both diseases are complex. The patients’ subjective experiences with diseases and medications are of utmost importance in pharmaceutical care practice. This study aimed to understand the subjective medication experience of TB and HIV/AIDS coinfected patients. The study was based on descriptive research of a qualitative and quantitative nature using data collected during pharmaceutical care appointments and from medical records from September 2015 to December 2016 at a tertiary infectious diseases referral hospital in Southeastern Brazil. Data from 81 patients were analyzed. Regarding patient subjective medication experience, the following responses to the quantitative questionnaire were most frequent: preference for a route of administration (12.4%) and for non-pharmacological therapy (50.6%); concerns about price (11.1%) and adverse effects (18.5%); and association of a worsening of their health status with a change in medication dosage (23.5%). In the thematic analysis, adversity and socially constructed aspects were more prominent. Resolvability, associated with the patient’s understanding of relief from signs and symptoms and health recovery, was observed; however, feelings of ambivalence permeated the other aspects, hence leading to treatment abandonment. The evaluation of patient medication experience can be a path to understanding and intervening in the phenomenon of treatment abandonment among TB and HIV/AIDS coinfected individuals.

## 1. Introduction

The stigma related to tuberculosis (TB) and human immunodeficiency virus/acquired immunodeficiency syndrome (HIV/AIDS) coinfection and its complex pharmacotherapy with a high number of drugs can cause the groups most affected by these diseases to be silenced, who are usually individuals from disadvantaged population strata, marked by poverty and violation of the fundamental right to live with dignity [1]. In this regard, patients should be aware of the need to complete the TB treatment regimen to obtain a cure, and should adhere to antiretroviral drugs to increase survival and prevent transmission of either TB or HIV/AIDS to other people. However, other aspects related to the patients themselves may make treatment adherence difficult [2]. Several studies on the factors associated with TB treatment abandonment have been described, revealing a serious problem in Brazil [1,3,4], especially when such abandonment takes place among patients living with TB and HIV/AIDS [1,3]. Among the associated factors is the large number of pills taken by these patients [4]. In this context, understanding their subjective medication experience may elucidate aspects related to patient outcomes.

The patient’s subjective experience, either with the disease or with the use of medications, is of utmost importance in pharmaceutical care practice [5]. This includes the patient’s preferences, feelings, concerns, beliefs and behaviors associated with the medications, whether based on his/her own use or on someone else’s use of such medications [6,7].

The concept of medication experience is defined as: “an experience of ambivalence and vulnerability in which the patient is actively engaged in an ongoing process or negotiation, which is pragmatic regarding the ways in which patients live and experience life, contextualized and nuanced within the social construction of their individual realities” [8].

Patients’ medication experience is at the center of pharmaceutical care practice and is the key to what makes it patient-centered [8]. In this process, health care professionals must assume a lucid and profound posture with a commitment to truly see the other, listening to patient subjectivity and abandoning the privileged position of a knowledge holder, thus giving the word to the individual [9]. The medication experience approach has been shown to be a useful way to prevent, identify and solve drug therapy problems, which impact on patients morbidity and mortality [10,11].

There are comprehensive studies on the topic of TB treatment abandonment, but few studies have been developed regarding the patient medication experience in the context of TB and HIV/AIDS coinfection [12]. It is necessary to elucidate how treatment abandonment is influenced by feelings and decision-making related to the medication experience. Therefore, this study aimed to understand the medication experience of TB/HIV coinfected patients and its relationship with TB treatment outcomes among coinfected individuals.

## 2. Methods

This study was structured according to the Standard for Reporting Qualitative Research (SRQR) [13].

The study is part of a larger project called Pharmaceutical care applied to patients living with TB and HIV/AIDS at a referral hospital, in Belo Horizonte, approved by the Research Ethics Committee of Federal University of Minas Gerais (CAAE: 23692713.3.0000.5149) and Hospital Eduardo de Menezes of the Hospital Foundation of the State of Minas Gerais State (CAAE: 23692713.2.3001.5124), where the study was conducted. Only information that addressed the medication experience of TB/HIV patients was selected for this study. In the period from September 2015 to December 2016, 81 patients under treatment for TB and HIV/AIDS were clinically followed-up by research pharmacists in a real-world study [14,15], using the theoretical and methodological framework of pharmaceutical care practice [5]. Outcomes were assessed while patients were followed-up in the hospital after starting TB treatment. Free and informed consent to participate in the study was obtained from all patients.

The study was based on descriptive research of a qualitative and quantitative nature and data were collected from interviews during clinical care and medical records. In the clinical interviewing process, the pharmacists used a tool [5] that helps systematize the collection and recording of such subjective data. The approximate duration of each interview was 60 min. The topics used in the interview are numbered in the questionnaire in Figure 1 and the alternatives were used to help in the development of the themes. In the questionnaire, there was an open field in which information was written according to the patients’ reports.

The data from the pharmacists’ notes on the assisted patients’ medication experience were submitted to thematic analysis, which involved data pre-analysis, data exploration and results data processing [16,17,18]. A holistic approach was used, which involved reading and rereading the data and pinpointing the initial ideas for familiarization. This was followed by coding relevant characteristics, grouping the resulting codes and checking whether the themes worked against the coded extracts and the entire data set, thus generating a thematic map of analysis. Further analysis was performed to refine the specifics of each theme, and finally vivid and compelling examples of the data extracts were selected [17,18]. The entire process was performed using the qualitative data analysis software N-vivo 10, Belo Horizonte, Brazil. The themes previously associated with the medication experience in the scientific literature [8,19] were used as a theoretical framework for the thematic analysis and for the discussion of the results.

The collected quantitative data were analyzed descriptively by means of a semi-structured questionnaire, which included the description of the study population and frequency distributions of the categorical variables.

The resulting qualitative and quantitative data are presented interleaved throughout the results section, since there is convergence between the themes covered.

## 3. Results

### 3.1. Characterization of the Study Participants

Of the total number of patients included (*n* = 81), 62 (77%) were male, 39 (48%) were older than 40 years and 59 (73%) had up to eight years of schooling. Patients with important social vulnerabilities were identified, such as: homeless people (*n* = 7, 8%), deprived of freedom (*n* = 2, 2%), and drug users (*n* = 18, 22%). In terms of clinical evaluation, 29 (35.8%) achieved TB cure, 28 (34.6%) were referred to another health facility, 14 (17%) died, five (6.2%) abandoned the treatment, and four (4.9%) had a change in their TB diagnosis.

### 3.2. Medication Experience

In the thematic analysis, the themes that emerged encompassed adversity, socially constructed aspects, resolvability and ambivalence, which are patient-related aspects that can play a role in treatment abandonment, which culminated in hospitalization and development of AIDS-defining diseases, such as TB.

The abandonment-related characteristic identified was indifference to treatment, regarded as a denial of the disease situation which culminated in non-adherence to treatment, as if the patient had ignored his or her clinical condition. Many of the followed-up patients denied the diagnosis of HIV/AIDS and chose not to treat it and/or ignored the diagnosis for several reasons. Patients’ statements regarding treatment abandonment are described below (Table 1): 

In terms of medication experience, the most frequent responses to the quantitative questionnaire were: preference for a route of administration (12.4%) and for non-pharmacological therapy (50.6%); concerns about price (11.1%) and adverse effects (18.5%); and association of a worsening in their state of health with a change in medication dosage (23.5%), as shown in Table 2.

Most patients denied that they objected to using medication (55.6%). However, the main reasons among those who reported objection were route of administration (12.4%) and price (11.1%).

Difficulty in having access to their medications was often reported referring to medication unavailability at the primary healthcare center, which might be the cause of nonadherence.

“Sometimes there is no medicine at the health center, and you have to buy it; (…) it’s not cheap” (P12).

The pharmacotherapy complexity was also a theme frequently reported during interviews. Below are some statements about the number of pills: “just in the morning, it’s more than 12” (P29). Patient feels uncomfortable with the size of the pill used to treat TB (P29). Patient had shingles during follow-up and at one point needed to use 39 pills a day (P4).

Adverse drug reactions (ADRs) may be associated with the pharmacotherapy complexity, since excessive polypharmacy, i.e., the use of 10 or more medications, was a fairly frequent feature among these patients (*n* = 69, 85%).

Although patients reported no association of culture or religion with medication use, religious coping was reported as a measure that helps with adherence to therapy. Some patients reported that religious faith supports treatment adherence: “Faith helps me take the medicines.” (P52).

Both current adverse effects and those experienced in the past concern patients. Although most patients did not report concern about adverse effects (56.8%), the analysis of such data helped understanding their experience of facing adversity. In the statements below (Table 3), patients refer to the impression of being subjected to experiments. The lack of knowledge and guidance about a drug that can cause an adverse reaction brought about a feeling of inferiority in these patients. Adverse reactions can interfere with work activities and impair adherence to treatment.

The theme of socially constructed aspects refers to the stigmas related to TB and HIV/AIDS diseases and the use of medication for their treatment. The patient’s realization of having to use medication for life and follow a recommended regimen also makes medication a symbol of dependence [8,20].

Most of the patients did not show any concerns regarding dependence on medications. Regarding generic, reference and similar drugs, the patients resorted to common sense to report on drug production in the country, pointing to social objects in this realm (Table 4).

The analysis showed that non-pharmacological therapy is preferred by patients; however, in the face of a chronic disease that cannot be cured but can be controlled, the statements of some patients brought about aspects related to resolvability. These are patients who accept the treatment in a practical and solution-oriented way (Table 5): 

Drug regimen changes and new medication additions meant for most patients a necessary adjustment, and for others a worsening in their health status, as if it was “a response from the body” (P34).

The patients’ accounts below reflect feelings of threat, imprisonment and anticipatory anxiety related to the treatment, one that is recognized as necessary but to which they adhere with fear and concern. Patients’ ambivalence towards medication use may contribute to the phenomenon of treatment abandonment (Table 6).

## 4. Discussion

The thematic analysis allowed observation of the fact that the patients’ statements noted in the medical records are consistent with findings of related studies describing patient experience with treatment abandonment, adversity, socially constructed aspects, resolvability and ambivalence [2,3,7,8].

Treatment abandonment is a widely explored topic in the literature, and understanding the reasons why patients abandon their treatments is paramount for the field of pharmaceutical care and for public policy making on TB/HIV coinfection [1,3]. This is an unfavorable outcome that can contribute to patient death. By definition, TB treatment abandonment occurs when the patient fails to attend the service appointments within a period of 30 consecutive days after the scheduled return date [21]. HIV/AIDS treatment abandonment therefore occurs when the patient, for more than 100 days, continues not going to the drug dispensing unit to withdraw antiretroviral drugs [22]. Issues such as social vulnerability, lack of family support, regret and use of alcohol and other drugs emerged in the patients’ statements regarding treatment abandonment. In the context of TB and HIV/AIDS coinfection, illness occurs in scenarios of extreme social vulnerability. Aspects related to low level of education and lack of formal employment permeate the individuals’ perception of the health service and interfere with their adherence to drug treatment [23,24]. Because this study was conducted at a tertiary referral hospital, some patients are followed-up only during hospitalization and are then referred to services closer to their homes, a fact that explains the high referral rate. However, it was also observed that a large proportion of the patients died, and that the TB cure rate is well below the national average of 53.4% in 2018 [25]. From these outcomes, it is noted that the pharmacotherapeutic monitoring of TB/HIV coinfected patients is a major challenge, underlying several situations related to the medication use.

The experience of adversity has also been described in qualitative studies addressing TB and HIV/AIDS co-infection and it comes into view when the actual adverse reactions or fear of them outweigh the perceived benefits of the medications themselves [7]. Drug-induced hepatitis is a common ADR among these patients [3]. From the patients’ perspective, the fact of having the treatment suspended, being given alternative regimen and then reintroduced to TB drugs one by one, as recommended by the protocol in the face of hepatotoxicity ADRs [21], gives them a feeling of being under an experiment. The fact of not knowing the adverse reactions brings about a feeling of being submissive to the treatment, with no option, hence impairing their autonomy. This autonomy restriction is a risk factor for some major worsening in the patient’s health [26]. The experience of adversity aroused negative feelings in the patients, therefore the practice of valuing the patients’ autonomy regarding their therapeutic choices and procedures to be followed should be encouraged. This characterizes the model in which patients and health professionals are seen as co-responsible for the treatment [27].

The medication experience is shaped by shared and accepted social and cultural ideas that define individual realities [28]. These are historical, social and cultural influences that can change over time, characterizing socially constructed aspects [8]. The feelings resulting from using medications in front of others can vary from individual to individual, and may involve feelings of embarrassment and personal failure. Drug dependence influences a person’s sense of self and his/her independence and functioning. This reinforces the perception that they are no longer the person they used to be. The unremitting nature of a chronic illness then becomes a burden, placing the patient in a passive position that leads to a sense of dependence [6]. TB and HIV/AIDS are two diseases that are closely related in that they are chronic and therefore require specific and long-term care [12]. Some patients in this study had the perception of generic drugs as posing more risks than reference drugs, believing that generic drugs may cause more side effects and may be less effective for this treatment in their country, disregarding the fact that the generic drug strategy represents an advancement in terms of access to medicines in Brazil [29]. Common sense models are idiosyncratic and are updated according to previous illness experiences, socio-cultural environment influences, namely mass communications, in addition to information exchanged with elements of the social support network or even with health professionals [29].

The experience of resolvability involves using medications sensibly and realistically, described as pragmatic and concrete decisions [8]. It is perceived by patients when the medication solves the problems caused by the disease and when it does not cause adverse reactions that may hinder their daily life [7]. There are patients who use up to ten medications without problems and others who recognize the chronicity of the disease in a positive feeling of continuity of life. In this concept, patients understand their medication experience as positive due to their bodily experiences, since the medication brings relief and allows them to recover their healthy body [6]. In pharmaceutical care practice, adherence to pharmacotherapy, when it is indicated, effective and safe, is a part of patient recovery or of a healing journey that involves struggle and requires the construction of skills in the face of a diagnosis of stigmatized diseases [8,30]. Some patients use faith as a source of strength, comfort and hope for personal empowerment, to fight the disease and use the medications [31].

The experience of ambivalence is the state of a person who has simultaneous and contradictory attitudes or feelings towards something [8]. The ambivalence between the demand and the subject who adheres to a treatment and at the same time is uneasy with the use of the medication is a logic that must be taken into consideration. Despite some of the interviewed patients being assiduous about appointments and treatment, this study noted through their statements the feelings of fear, anxiety, resistance, and the notion that they should avoid medication. In this case, medication is associated with something bad, and negative feelings occur even before the use of that medication [6]. This is a feeling of anticipatory anxiety in which the medication is administered with fear by the patient. The individual becomes anxious just at the remembrance of what generates his/her phobia. A phobic person constantly avoids contact with the element that causes his or her discomfort [32]. As the participants accounted their medication experience, it became more evident how fluid and dynamic such experiences can be [33]. Therefore, the concept of ambivalence applies well to these patients with contradictory thoughts, since, given the complexity of TB and HIV/AIDS treatment, doubt and feelings of ambiguity arise. These feelings can interfere with the achievement of therapeutic goals.

When interviewed, these patients were not aware of everything that could be understood from their statements about their experience and that of their contemporaries in society [18]. Their accounts contributed to the understanding of how their subjective experience interferes with their health outcomes and can help health care professionals reflect on their situationality, increasing reflexivity and helping them look more critically at themselves as professionals and human beings. This makes room for greater awareness of how they relate to patients and other health professionals [34]. Ongoing reflection and documentation of the medication experience are crucial to continue advancing patient-centered pharmacy practice [33].

Having a positive influence on people’s feelings about their treatments so that they become more aware, and on their social situation is a great challenge for health professionals. Housing situations, violence, fear, anxiety and regret at abandoning the treatment previously are all raised as themes related to the medication experience. The directly observed treatment strategy (DOT), via dialogic action, is an important way to help patients in the awareness process [21], since multi-professional action in this care can change the perceptions and attitudes of individuals in relation to the use of medicines, in addition to guaranteeing assistance regarding other obstacles to treatment, so common in this population. These patients were treated in a specialized care service in Brazil and the integration with primary care, through DOT, is essential to achieve the effectiveness of the treatment, especially in larger cities, such as the place where the study was conducted. The expansion of DOT, which is already carried out in primary care, and of social protection programs can guarantee better results for TB indicators [35].

The use of the tool with open-ended and close-ended questions proved useful for patient-centered pharmaceutical care as it directs what one wants to know about patients´ medication experiences for the improvement of care, as a guide for the appointment, and hence not overlooking the experiences worthy of attention that emerge [33]. In this regard, research on the medication experience of TB/HIV coinfected patients using the mixed method brings important contributions to increasing knowledge about the use of medication experience in clinical practice.

The strength of this study is its proposal to evaluate the medication experience of TB/HIV coinfected patients, as this is an incipient research topic in the literature. However, there are limitations regarding data collection, given the fact that patients´ accounts and statements were taken from pharmacists´ notes in medical records. Because of this, it was not possible to obtain all the literal answers from the patients and, therefore, pharmacists´ reports were added. Qualitative studies with in-depth interviews, with appropriate recording techniques, are necessary for a better understanding of the medication experience among the population coinfected with TB and HIV/AIDS.

## 5. Conclusions

The findings of this study point to the need for professional action in a dialogical manner with patients in their therapeutic process, considering that their medication experience can be either positive or negative, but is capable of being modified over time. Assessing patient medication experience can be a path to understanding and intervening in the phenomenon of treatment abandonment, as it seems to lie at the root of TB and HIV/AIDS coinfection-related outcomes. From this dialogical action, health professionals can act in the education and transformation of patients and the community. Strategies such as DOT are important in this process, as well as the interaction with primary health care. Therefore, actions are needed to strengthen inclusion policies for the protection of rights, such as social programs and strategies aimed at fighting discrimination and prejudice. Policies should be created to eliminate barriers and expand access to public goods and services, of which many do not often reach those populations that drop out of treatment.

## Figures and Tables

**Figure 1 ijerph-19-15153-f001:**
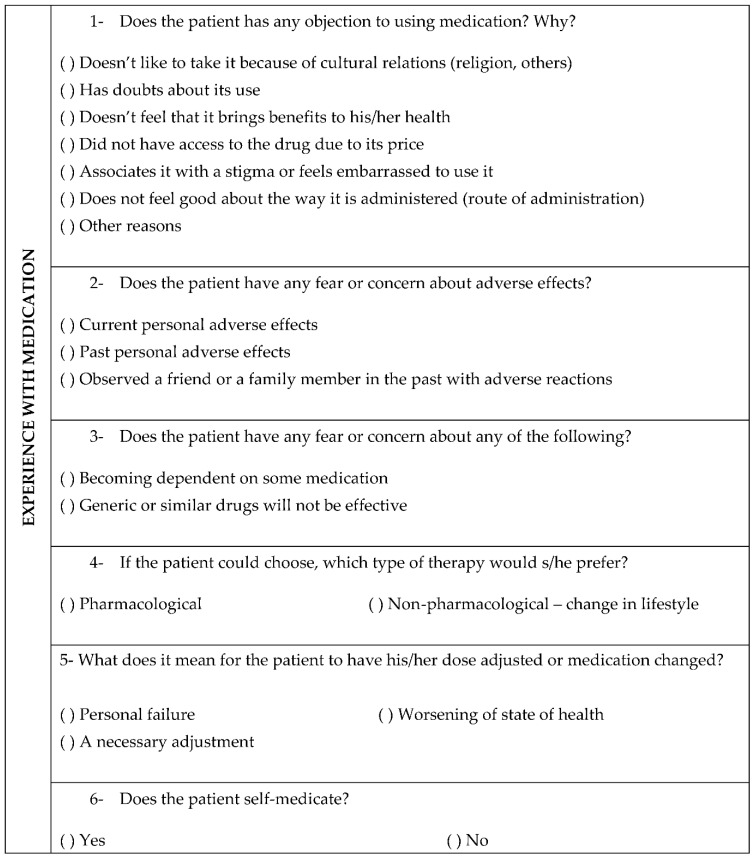
Instrument used to collect subjective patient data.

**Table 1 ijerph-19-15153-t001:** Treatment abandonment description.

Theme 1: Treatment Abandonment
Patient states that if s/he had not abandoned the treatment, s/he would not have been hospitalized (P25).
Patient associates the dose adjustment or inclusion of a new drug in the therapy as a personal failure because s/he had experienced treatment failure due to non-adherence in the past. Patient reports non-adherence to antiretroviral therapy because of a personal preference not to use the medications (P68).
“I didn’t value the medication”. S/He is an alcoholic and a smoker, lives alone with no family support “in a little shed with two rooms, a bathroom and a washing tank area”. Patient has a history of poor treatment adherence and multiple dropouts. Patient has no option to pick up medications elsewhere when they are not available at the primary healthcare unit (P51).
Young patient with a history of sexual violence at the age of 13 (probable source of HIV/AIDS transmission). Patient expressed to an aunt no intention to be treated and a will to die, thus “throwing in the towel”. Patient did not continue the treatment, even though the medications were available. The follow-up nursing technician reported that this patient refuses the medications and sometimes throws them away and feels “very confused” (P3).

**Table 2 ijerph-19-15153-t002:** Description of the experience with medication use among patients with tuberculosis and HIV/AIDS at Hospital Eduardo de Menezes, city of Belo Horizonte, Brazil (*n* = 81).

Variable	*n*	%
**Does the patient have any objection to using the medication?**		
Does not like to take it for cultural reasons (religion, others)	0	0.0
Has doubts regarding its use	7	8.6
Does not feel that it brings benefits to his/her health	1	1.2
Did not have access to the drug due to its price	9	11.1
Associates it with a stigma or feels embarrassed to use it	4	4.9
Does not feel comfortable with the way it is administered (route of administration)	10	12.4
Price and route of administration	3	3.7
Has no objection	45	55.6
Other reasons	1	1.2
Did not answer	1	1.2
**Is the patient afraid or concerned about adverse effects?**		
Current personal adverse effects	15	18.5
Past personal adverse effects	15	18.5
Observed a friend or a family member in the past with adverse reactions	2	2.4
Current and past personal adverse effects	2	2.4
No fear or concern about adverse effects	46	56.7
Did not answer	1	1.2
**Is the patient afraid or concerned about any of these below?**		
Becoming dependent on some medication	16	19.7
Becoming dependent on some medication and generic or similar drugs are not effective	4	4.9
Generic or similar drugs are not effective	8	9.9
No fear or concern about these aspects	52	64.2
Did not answer	1	1.2
**If the patient could choose, which type of therapy would s/he prefer?**		
Pharmacological	24	29.6
Non-pharmacological—change in lifestyle	41	50.6
Pharmacological and non-pharmacological	8	9.9
None	3	3.7
Did not answer or does not know	5	6.2
**What does it mean for the patient to have his/her dose adjusted or medication changed?**		
Personal failure	5	6.2
Worsening in state of health	19	23.5
Personal failure and worsening in state of health	1	1.2
A necessary adjustment	50	61.7
Did not answer or does not know	6	7.4
**Does the patient self-medicate?**		
No	29	35.8
Yes	48	59.3
Did not answer	4	4.9

**Table 3 ijerph-19-15153-t003:** Adversity description.

Theme 2: Adversity
“They didn’t tell me the name (of the medicine), as if I was a doormat… experimenting the shot [in me]” (P15).
Patient stated that s/he felt like a “guinea pig” with the withdrawal and introduction of medications (P2).
Patient stated that when s/he took the medication at night s/he had neuropsychiatric effects, and this prevented him/her from working. S/He was a machine operator and was off the antiretroviral drugs for three years. When s/he returned to the hospital, s/he had AIDS and tuberculosis. S/He claims to be afraid of the adverse effects s/he had in the past with a yellow pill that caused him/her insomnia (P37).

**Table 4 ijerph-19-15153-t004:** Socially constructed aspects description.

Theme 3: Socially Constructed Aspects
“I feel embarrassed to take (the medicines) in front of other people” (P51).
“It’s hard for me to take home so many medicines” (P68).
“I’m afraid of becoming dependent on medications, even though I have already become (drug cocktail)” (P76).
Patient states that s/he is afraid of becoming dependent on any medication and that s/he is “anti-drug” (P34).
Patient was healthy before (…) and vented “Now I’ll have to be dependent on medication.” (P22)
Patient claims to be afraid or concerned that generic or similar drugs are not effective, because “in Brazil (they) don’t believe in generic drugs; (they) always prefer the original” (P40).

**Table 5 ijerph-19-15153-t005:** Resolvability description.

Theme 4: Resolvability
Patient states that s/he has no problems with using medications and states that s/he “takes up to ten pills at once.” Has observed adverse drug reactions in his/her spouse but reports no fear or concern. S/He prefers a change in lifestyle to taking medications but understands that antiretroviral treatment is of continuous use (P12).
Patient states, regarding antiretrovirals, that “will never stop [taking them]; only when there is a cure” (P10).

**Table 6 ijerph-19-15153-t006:** Ambivalence description.

Theme 5: Ambivalence
“I find it strange to use this much medication, I feel hostage to using these medicines. I avoid getting drugs into my body as much as possible” (P17).
Patient stated that the medications s/he uses are very strong and that s/he gets worried about getting worse with the use of these drugs (P23).
S/He also claimed that the medications bring benefits, but “s/he doesn’t like to take them very much” (P5).
In relation to the medications, patient states that “s/he takes them with fear indeed” (P53).
S/He reported concern about using the medications, since s/he did not take any before being hospitalized and knows that when s/he is discharged s/he will have to use many (P20).

## Data Availability

Data were obtained from the SUS of Belo Horizonte and are available from the coordinator researcher.

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
