# Peer review of "The Medication Experience of TB/HIV Coinfected Patients: Qualitative Study"

_ijerph, 2022, doi:10.3390/ijerph192215153_

Round 1
Reviewer 1 Report
This is a good study evaluating the attitude and perception of TB/HIV patients and the potential therapeutic barriers. Several comments need to be addressed.
1. Title: Please indicate that this is a qualitative study.
2. Abstract: Line 12: Please spell out HIV/AIDS
Introduction:
3. Lines 31-39 are part of the instructions to authors. Please delete.
4. Line 44: Please spell out TB.
5. Line 45: Please spell out HIV/AIDS.
6. Line 57: The statement doesn’t make sense. Please rephrase or remove.
Methods:
7. Line 85: Please provide more details regarding the interview process in brief.
8. It looks like the tool presented in Figure 1 contains only close-ended questions. But since the results section includes some descriptive responses from the participants, do the authors have the open-ended questions used by the interviewing pharmacists (or at least the topics of these questions) that aided the development of the themes for thematic analysis?
9. Do the authors know the approximate duration of each interview? If so, please add it to the methods.
10. Line 108: Please use the past tense (change “will be” to “were”).
11. Question 4 in Figure 1 doesn’t seem to be applicable in the setting of TB and HIV. Please justify having this question.
Results:
12. Line 115: Correct "evolution" to "evaluation"
13. Tables 1, 3, 4, and 5: It might be better to be consistent in including the statements in the participants' own words instead of reporting some verbatim and some being rephrased. This would also make it easier to report responses from patients with different unclear gender preference (e.g., P5, P12, P20, P37, P51, and P68). If this is not available, then please add the lack of verbatim responses to the limitation that the data were collected from the medical records.
Discussion:
14. I suggest adding a point about DOT (direct observed therapy). Is this used in Brazil (or in the city where the study was conducted)? Was any of the participants part of this program? How having a DOT may change patients' perceptions and attitudes toward the pill burden and other obstacles (i.e., advantages of implementing a DOT program)?
Conclusion:
15. Please expand the conclusion and your recommendation to the healthcare team caring for TB/HIV coinfected patients. Perhaps also add a recommendation regarding establishing a DOT program and address the stakeholders who would be responsible in establishing and executing such program.
Author Response
November, 08th 2022
Dear Reviewer,
On the MDPI’s website, please find a revised version of our manuscript, entitled “The Medication Experience of TB/HIV Coinfected Patients: Qualitative Study”, protocol number ijerph-1974492. In this version, we have considered the comments raised by the three reviewers, and we have revised our paper accordingly. A point-by-point answer is also provided herein and highlited in red.
We would like to thank the reviewers for their constructive comments, which we feel have contributed greatly to improving the quality of our manuscript. We hope that this revised version will be more comprehensive and that it will be considered for publication in the International Journal of Environmental Research and Public Health.
Sincerely,
Natália Helena de Resende (corresponding author)
The incorporated revisions are listed below.
This is a good study evaluating the attitude and perception of TB/HIV patients and the potential therapeutic barriers. Several comments need to be addressed.
- Title: Please indicate that this is a qualitative study.
This information was inserted as requested.
- Abstract: Line 12: Please spell out HIV/AIDS
The meaning of the initials HIV/AIDS has been written as requested.
Introduction:
- Lines 31-39 are part of the instructions to authors. Please delete.
The instructions have been deleted.
- Line 44: Please spell out TB.
The meaning of the initials TB has been written as requested.
- Line 45: Please spell out HIV/AIDS.
The meaning of the initials HIV/AIDS has been written as requested.
- Line 57: The statement doesn’t make sense. Please rephrase or remove.
The statement has been removed as requested.
Methods:
- Line 85: Please provide more details regarding the interview process in brief.
The interview process brief was added in the methodology.
- It looks like the tool presented in Figure 1 contains only close-ended questions. But since the results section includes some descriptive responses from the participants, do the authors have the open-ended questions used by the interviewing pharmacists (or at least the topics of these questions) that aided the development of the themes for thematic analysis?
This information was added in the methodology. The topics used in the interview were numbered in the questionnaire in Figure 1 and the alternatives were used to help in the development of the themes. The questionnaire was an open field which the information was recorded according to the patients' reports.
- Do the authors know the approximate duration of each interview? If so, please add it to the methods.
This information was added in the methods: “The approximate duration of each interview was 60 minutes.”
- Line 108: Please use the past tense (change “will be” to “were”).
The verb tense has been changed as requested.
- Question 4 in Figure 1 doesn’t seem to be applicable in the setting of TB and HIV. Please justify having this question.
As described in the text, this study is part of a larger project entitled “Pharmaceutical care applied to patients living with TB and HIV/AIDS at a referral hospital, in Belo Horizonte” and because of this the questionnaire was wider. Only the excerpts that address the medication experience of TB/HIV patients were selected. This excerpt was added to the text in the methods.
Results:
- Line 115: Correct "evolution" to "evaluation"
The word has been corrected as requested.
- Tables 1, 3, 4, and 5: It might be better to be consistent in including the statements in the participants' own words instead of reporting some verbatim and some being rephrased. This would also make it easier to report responses from patients with different unclear gender preference (e.g., P5, P12, P20, P37, P51, and P68). If this is not available, then please add the lack of verbatim responses to the limitation that the data were collected from the medical records.
The lack of literal responses was added to the discussion as limitations of the study. Not all patients presented literal statements and the interviewer's report was added. Research using the interview recording technique should be carried out to facilitate obtaining literal answers.
Discussion:
- I suggest adding a point about DOT (direct observed therapy). Is this used in Brazil (or in the city where the study was conducted)? Was any of the participants part of this program? How having a DOT may change patients' perceptions and attitudes toward the pill burden and other obstacles (i.e., advantages of implementing a DOT program)?
We added a paragraph in the discussion about the aspects asked. The study was carried out in a specialized HIV/AIDS care service and patients are followed up through the DOT program in primary health care. The discussion highlighted the importance of integrating the two services.
Having a positive influence on people's feelings about treatment so that they become aware and on their social situation is a great challenge for health professionals. The housing situation, violence, fear, anxiety and regret of abandoning treatment previously raised themes related to the medication experience. The directly observed treatment strategy (DOT) is an important way to help patients in the awareness process, since the multidisciplinary action in this care can change the perceptions and attitudes of individuals in relation to the use of medicines. In addition to guaranteeing assistance in other obstacles in the treatment common in this population from a dialogic action. These patients were treated in a specialized care service in Brazil and the integration with primary care, through DOT, is essential to achieve the effectiveness of the treatment, especially in larger cities, such as the place where the study was developed. The expansion of DOT, which is already carried out in primary care, and of social protection programs can guarantee better results for TB indicators.
Conclusion:
- Please expand the conclusion and your recommendation to the healthcare team caring for TB/HIV coinfected patients. Perhaps also add a recommendation regarding establishing a DOT program and address the stakeholders who would be responsible in establishing and executing such program.
An excerpt was added to the conclusion with a recommendation to healthcare professionals who care for these patients as requested.
From dialogical action, health professionals can act in the education and transformation of patients and the community. Strategies such as DOT are important in this process, as well as the interaction with the level of primary health care. Therefore, actions are needed to strengthen the increase in inclusion policies for the protection of rights, such as social programs and strategies aimed at fighting discrimination and prejudice in order to eliminate barriers and expand access to public goods and services, which many often do not reach those populations that drop out of treatment.
Submission Date
30 September 2022
Date of this review
11 Oct 2022 11:34:36

Reviewer 2 Report
The manuscript demonstrated the medication experience of patients with TB and HIV coinfection, where many factors resulted in treatment abandonment, eventually causing death of the patients. The study is comprehensive and well-discussed.
Only minor revisions needed as follows:
Page 1 line 31-39, the paragraph of “The introduction should…further details on references” should be deleted.
Page 5 Table 2, for variable “Does not like to take it for cultural reasons (re-ligion, others)” N should be 0, if % is 0.0. Don’t leave it blank.
Some minor format mistakes need to be addressed. For example, the last reference did not be numbered in Page 11, line 388.
Author Response
November, 08th 2022
Dear Reviewer,
On the MDPI’s website, please find a revised version of our manuscript, entitled “The Medication Experience of TB/HIV Coinfected Patients: Qualitative Study”, protocol number ijerph-1974492. In this version, we have considered the comments raised by the three reviewers, and we have revised our paper accordingly. A point-by-point answer is also provided herein and highlited in red.
We would like to thank the reviewers for their constructive comments, which we feel have contributed greatly to improving the quality of our manuscript. We hope that this revised version will be more comprehensive and that it will be considered for publication in the International Journal of Environmental Research and Public Health.
Sincerely,
Natália Helena de Resende (corresponding author)
The incorporated revisions are listed below.
The manuscript demonstrated the medication experience of patients with TB and HIV coinfection, where many factors resulted in treatment abandonment, eventually causing death of the patients. The study is comprehensive and well-discussed.
Only minor revisions needed as follows:
Page 1 line 31-39, the paragraph of “The introduction should…further details on references” should be deleted.
This paragraph has been excluded from the introduction.
Page 5 Table 2, for variable “Does not like to take it for cultural reasons (re-ligion, others)” N should be 0, if % is 0.0. Don’t leave it blank.
The number 0 has been added to the table as requested.
Some minor format mistakes need to be addressed. For example, the last reference did not be numbered in Page 11, line 388.
The reference was numbered in the text as requested.
Submission Date
30 September 2022
Date of this review
11 Oct 2022 18:52:38

Reviewer 3 Report
Figure 1. The questions are quite vague and expressed suggestively!
Question 1: The first question should not read “Why” (suggesting that naturally there are any objections) but: “Given that the patient has (not “have”) any objection to using medication, why?”
Did the patients previously experience any drug therapy? If not, how should the “doubt about its use”?
Question 3: What has fear of drug-dependence to do with effectiveness?
Question 4: It should be clear that TB and/or HIV cannot be treated by changes in lifestyle. Thus, it is entirely superfluous to ask for such an option!
Line 88: “ A holistic approach was used, which involved reading and rereading the data and pinpointing the initial ideas, for familiarization.” What´s the scientific basis for this procedure? (reading…rereading…pinpointing)
Line 101, 3.1. Characterization of the study participants: How many of the patients were currently under treatment or had a prior treatment? Does “referred to another health facility” mean that those patients had already started treatment? What does “had a change in TB diagnosis” mean? Did the persons have active TBs or not?
Line 133 ff “Some patients reported that they regretted having abandoned the treatment. Among them, the most recurrent accounts and statements regarded absenteeism from clinic apppointments, multiple abandonments, and non-adherence to treatment, which culminated in hospitalization and development of AIDS-defining diseases, such as TB.”
and
166ff: “The theme of socially constructed aspects refers to the stigmas related to TB and HIV/AIDS diseases and the use of medication for their treatment. This represents an as sociation thereof with social symbols or objects. The patient’s realization of having to use medication for life and follow a recommended regimen also makes medication a symbol of dependence”
Such sentences do not contain useful, but are a conglomeration of empty phrases!
Line 303ff: “The findings of this study point to the need for professional action in a dialogical manner with patients in their therapeutic process, considering that their medication experience can be either positive or negative, however capable of being modified over time” See my comment above.
In my view a professional action is always required. The patients´ attitude on their medication, however, must be investigated in a scientifically structured procedure regarding the concrete access to treatment and concrete adverse effects of the used medicaments.
Author Response
November, 08th 2022
Dear Reviewer,
On the MDPI’s website, please find a revised version of our manuscript, entitled “The Medication Experience of TB/HIV Coinfected Patients: Qualitative Study”, protocol number ijerph-1974492. In this version, we have considered the comments raised by the three reviewers, and we have revised our paper accordingly. A point-by-point answer is also provided herein and highlited in red.
We would like to thank the reviewers for their constructive comments, which we feel have contributed greatly to improving the quality of our manuscript. We hope that this revised version will be more comprehensive and that it will be considered for publication in the International Journal of Environmental Research and Public Health.
Sincerely,
Natália Helena de Resende (corresponding author)
The incorporated revisions are listed below.
Figure 1. The questions are quite vague and expressed suggestively!
Question 1: The first question should not read “Why” (suggesting that naturally there are any objections) but: “Given that the patient has (not “have”) any objection to using medication, why?”
We consider the comment pertinent and that it is really important to ask more open and less suggestive questions, thinking from the perspective of qualitative research. The team based itself on the scientific literature in the area using the instrument in Figure 1, by the authors Cipolle, Strand and Morley (2012). The questionnaire was used in the context of a clinical consultation and it was possible to explore the topics listed in the questionnaire in a wide way, according to the patient's experience. That is, it was not used as a structured questionnaire, but as guiding topics for the interview with patients. This information was written clearer in the methodology: “The topics used in the interview were numbered in the questionnaire in Figure 1 and the alternatives were used to help in the development of the themes. In the questionnaire there was an open field in which the information was written according to the patients' reports.”
Did the patients previously experience any drug therapy? If not, how should the “doubt about its use”?
All the evaluated patients were patients on drug treatment coinfected with tuberculosis and HIV/AIDS, as explained in the methods: “81 patients under treatment for TB and HIV/AIDS were clinically followed by research pharmacists in a real-world study.”
Question 3: What has fear of drug-dependence to do with effectiveness?
The medication experience interferes with the patient's behavior towards the treatment, so the fear of becoming dependent on medication can make the patient avoid the use and abandon the treatment, which interferes with the effectiveness of the treatment. This explanation is expressed in the second paragraph of the introduction and in the fourth paragraph of the article's discussion.
Question 4: It should be clear that TB and/or HIV cannot be treated by changes in lifestyle. Thus, it is entirely superfluous to ask for such an option!
As described in the text, this study is part of a larger project entitled “Pharmaceutical care applied to patients living with TB and HIV/AIDS at a referral hospital, in Belo Horizonte” and because of this the questionnaire was wide. Only the information that address the medication experience of TB/HIV patients were selected. This phrase was added to the text in the methodology.
Line 88: “ A holistic approach was used, which involved reading and rereading the data and pinpointing the initial ideas, for familiarization.” What´s the scientific basis for this procedure? (reading…rereading…pinpointing)
These are the methodological steps of the thematic analysis described in the following references: Minayo (2014) and Braun and Clarke (2006). One of these references is by researcher Minayo, who is one of the most important references in qualitative health research in Brazil. Braun and Clarke are important researchers who describe thematic analysis in qualitative studies. We added a reference to a specific article by the data analysis author supporting our methodological choice. Available at: https://www.scielo.br/j/csc/a/39YW8sMQhNzG5NmpGBtNMFf/?lang=en.
Line 101, 3.1. Characterization of the study participants: How many of the patients were currently under treatment or had a prior treatment? Does “referred to another health facility” mean that those patients had already started treatment? What does “had a change in TB diagnosis” mean? Did the persons have active TBs or not?
At the time of the interview, all patients were being treated for tuberculosis, as described in the methodology: “81 patients under treatment for TB and HIV/AIDS were clinically followed by research pharmacists in a real-world study.” Transfer and diagnoses change are outcomes assessed after the interviews. The follow information was included in the text: “Outcomes were assessed while patients were followed up in the hospital after initiation of TB treatment”.
Line 133 ff “Some patients reported that they regretted having abandoned the treatment. Among them, the most recurrent accounts and statements regarded absenteeism from clinic apppointments, multiple abandonments, and non-adherence to treatment, which culminated in hospitalization and development of AIDS-defining diseases, such as TB.”
and
166ff: “The theme of socially constructed aspects refers to the stigmas related to TB and HIV/AIDS diseases and the use of medication for their treatment. This represents an as sociation thereof with social symbols or objects. The patient’s realization of having to use medication for life and follow a recommended regimen also makes medication a symbol of dependence”
Such sentences do not contain useful, but are a conglomeration of empty phrases!
The sentences were removed and reformulated as requested and marked in the text in red.
Line 303ff: “The findings of this study point to the need for professional action in a dialogical manner with patients in their therapeutic process, considering that their medication experience can be either positive or negative, however capable of being modified over time” See my comment above.
In my view a professional action is always required. The patients´ attitude on their medication, however, must be investigated in a scientifically structured procedure regarding the concrete access to treatment and concrete adverse effects of the used medicaments.
We understand the importance of quantitative studies that use structured and validated questionnaires for the investigation of adverse reactions, adherence to treatment, among others. Our study, however, uses another methodological approach that is also scientifically based (the qualitative approach of medication experience) as can be seen in the following publications:
- Conrad P. The meaning of medications. Another look at compliance. Soc. Sci. Med. 1985, 20, 29-37.
- Shoemaker, S.J.; Ramalho de Oliveira, D. Understanding the meaning of medications for patients: the medication experience. Pharm World Sci. 2008, 30, 86-91.
- Redmond, S.; Paterson, N.; Shoemaker-Hunt, S.J; Ramalho-de-Oliveira, D. Development, Testing and Results of a Patient Medication Experience Documentation Tool for Use in Comprehensive Medication Management Services. Pharmacy (Basel), 2019, 7, 16.
- Shoemaker, S.J.; Ramalho de Oliveira, D.; Alves M.; Ekstrand, M. The medication experience: Preliminary evidence of its value for patient education and counseling on chronic medications. Patient Education and Counseling, 2011, 83, 3, 443-450.
- Mohammed M.A.; Moles, R.J.; Chen, T.F. Medication-related burden and patients’ lived experience with medicine: a systematic review and metasynthesis of qualitative studies. BMJ Open, 2016, ;6:e010035.
- Ramalho de Oliveira, D.; Alves, M. Understanding the Patient’s Medication Experience: Collaboration for Better Outcomes. Brill, 2014, ISBN: 9781848882621 https://doi.org/10.1163/9781848882621_007
- Hillman, L.A.; Peden-McAlpine, C.; Ramalho-de-Oliveira, D.; Schommer, J.C. The Medication Experience: A Concept Analysis. Pharmacy (Basel). 2020, 9, 17.
- Nascimento, Y.A., Silva, L.D., Ramalho-de-Oliveira, D. Experiences with the daily use of medications among chronic hepatitis C patients. Res Social Adm Pharm. 2020, 16, 33-40.
- Nascimento, Y.A.; Ramalho-de-Oliveira, D. The Subjective Experience of Using Medications: What We Know and the Paths Forward. Pharmacy (Basel). 2021, 9, 50.
- Nascimento, Y.A., Filard, AFR, Abath AJ, Silva LD, Ramalho-de-Oliveira D. The phenomenology of Merleau-Ponty in investigations about medication use: constructing a methodological cascade. Rev Esc Enferm USP. 2017, 51:e03296.
The medication experience approach has been shown to be a useful way to prevent, identify and solve drug therapy problems, which impact on patients morbidity and mortality. This information was added to the article's introduction to make it clearer for the reader.
Submission Date
30 September 2022
Date of this review
28 Oct 2022 16:28:55

Round 2
Reviewer 3 Report
The authors did a lot of work to improve their paper. Indeed, some of my concerns have been clarified. All in all, the paper is worth to be published now.